# Advanced Detection and Therapeutic Monitoring of Atherosclerotic Plaque Using CD36-Targeted Lipid Core Probe

**DOI:** 10.3390/pharmaceutics17040444

**Published:** 2025-03-30

**Authors:** Tingting Gao, Siqi Gao, Maolin Qiao, Chuanlong Lu, Heng Wang, Hongjiu Zhang, Lizheng Li, Shule Wang, Ruijing Zhang, Honglin Dong

**Affiliations:** 1The Second Clinical Medical College, Shanxi Medical University, Taiyuan 030000, China; gaotingting@sxmu.edu.cn (T.G.); gaosiqi2@sxmu.edu.cn (S.G.); qiaomaolin@sxmu.edu.cn (M.Q.); 17200588255@163.com (C.L.); wangheng7015@mails.jlu.edu.cn (H.W.); zhjiu09@163.com (H.Z.); phoenixllz@163.com (L.L.); wangshulego@163.com (S.W.); 2Centre for Transplant and Renal Research, Westmead Institute for Medical Research, The University of Sydney, Sydney 201101, Australia

**Keywords:** atherosclerosis, lipid necrotic core, fluorescence imaging, CD36, targeting probe

## Abstract

**Background:** Atherosclerotic diseases, including coronary heart disease and cerebrovascular disease, are leading causes of morbidity and mortality worldwide. Atherosclerosis is a chronic vascular condition marked by the accumulation of lipid plaque within arterial walls. These plaques can become unstable and rupture, leading to thrombosis and subsequent cardiovascular events. Therefore, early identification of vulnerable plaque is critical for preventing such events. **Objectives:** This study aims to develop a novel imaging platform for atherosclerotic plaque by designing a molecular imaging probe based on fluorescent molecules that target lipid necrotic cores. The goal is to specifically detect high-risk plaque, enabling early diagnosis and intervention. **Methods:** Bioinformatic analysis and immunofluorescence were used to detect CD36 expression in human carotid plaque. CD36pep-ICG was synthesized using the Fmoc solid-phase peptide method. A series of experiments was conducted to characterize the probe’s properties. To assess imaging performance, probe concentration gradients were tested using FLI equipment. Ex vivo imaging was performed on atherosclerotic mice and treatment models to evaluate the probe’s targeting ability and effectiveness in monitoring disease progression. **Results:** The CD36 expression was significantly elevated in the core of plaque compared to distal regions. The CD36pep-ICG probe, specifically designed to target lipids, was successfully synthesized and exhibited excellent fluorescence properties. In animal models, FLI imaging demonstrated that the CD36pep-ICG probe selectively accumulated in atherosclerotic plaque, enabling precise plaque detection. Moreover, the probe was used to monitor the therapeutic efficacy of anti-atherosclerotic drugs. **Conclusions:** The CD36pep-ICG probe developed in this study is an effective molecular imaging tool for the specific identification of vulnerable atherosclerotic plaque, offering a novel approach for early diagnosis and treatment. Additionally, the probe shows promise in tracking the therapeutic effects of the drug, potentially advancing the precision treatment of cardiovascular diseases.

## 1. Introduction

Atherosclerotic diseases, particularly coronary heart disease and cerebrovascular disease, are major contributors to global morbidity and mortality [1]. According to the World Health Organization (WHO), cardio-cerebral vascular diseases remain the leading cause of death worldwide, responsible for nearly one-third of all fatalities [2]. Atherosclerosis, a chronic vascular disorder resulting from dysregulated lipid metabolism, is characterized by the progressive buildup of lipid plaque within arterial walls [3]. In its early stages, atherosclerosis is often asymptomatic; however, as the disease advances, plaque becomes unstable and ruptures, leading to thrombosis and life-threatening cardiovascular events [4]. Therefore, the early and precise identification of vulnerable plaque is essential for timely and effective treatment strategies to prevent fatal cardiovascular incidents.

Highly selective and efficient targeting of atherosclerosis is critical for its precise detection [5]. Acute cardiovascular events are predominantly caused by the rupture or erosion of unstable, vulnerable plaque [6,7]. These plaques are characterized by a large necrotic lipid core, a thin fibrous cap, and significant infiltration of inflammatory cells [8]. The expansion of the necrotic lipid core increases plaque volume and reduces stability, ultimately leading to plaque rupture, thrombosis, and the onset of cardiovascular or cerebrovascular events [9].

Lipid-lowering therapies, such as statins, can improve plaque pathology. Statins, a widely prescribed class of cholesterol-lowering drugs, facilitate the transformation of lipid-rich atherosclerotic plaque into calcified plaque, a process known as “calcific densification” [10,11]. Calcified plaques are generally considered a more stable stage of atherosclerotic lesions, and promoting calcific densification can help reduce the risk of atherosclerosis progression.

Currently, the identification of unstable and vulnerable plaques remains a significant challenge. The lipid necrotic core, a key pathological feature of vulnerable plaque, comprises extracellular lipids, foam cells, and cell fragments that accumulate within the arterial intima [12]. CD36, a membrane surface protein, mediates the uptake of oxidized low-density lipoprotein (ox-LDL), a critical step in foam cell formation [13]. This process is considered an early event in the development of atherosclerotic plaque and a marker of the early progression of vulnerable plaque [14,15]. Molecular imaging targeting CD36 shows promise for the early detection of vulnerable atherosclerotic plaque [16,17]. Identifying the primary pathological characteristics of these plaques is expected to provide valuable diagnostic and therapeutic insights for clinical applications.

The imaging detection of vulnerable plaque primarily focuses on vascular morphology, with various imaging techniques playing a crucial role in the risk stratification of atherosclerotic plaque [18]. Molecular imaging methods, including ultrasound, computed tomography (CT), and magnetic resonance imaging (MRI) are actively being investigated for their potential in detecting atherosclerotic plaque [19]. However, while these techniques show promise, none are yet widely adopted in standard clinical practice. Each technique has specific limitations. Ultrasound provides the advantages of low-cost and real-time imaging but is constrained by shallow tissue penetration and operator dependency, making results heavily reliant on the technician’s expertise [20,21]. Similarly, MRI, though effective, requires a long acquisition time and is costly, necessitating substantial patient cooperation to avoid image blurring and artifacts. Coronary CT angiography (CCTA) is considered to be highly accurate in detecting low-density necrosis and lipid cores within plaque, as demonstrated by studies such as the SCOT-HEART trial [22,23]. However, it is limited by radiation exposure and low soft tissue contrast agents [24]. Notably, early-stage atherosclerosis can be reversed with pharmacological intervention, whereas advanced disease cannot. Thus, there is an urgent need for advanced imaging technologies to enable the early identification and prognosis assessment of atherosclerosis.

Molecular imaging, which focuses on biological processes, shows considerable potential for assessing vulnerable plaque. Molecular imaging uses targeted contrast agents or radioactive tracers to visualize and characterize biological processes at the organ, cellular, and subcellular levels [25]. An ideal platform for atherosclerotic plaque imaging requires highly sensitive imaging techniques and specifically targeted markers to identify high-risk, vulnerable plaque [26]. Fluorescence imaging (FLI) is particularly promising due to its high sensitivity, excellent spatial resolution, rapid acquisition, and efficient image processing [27]. It allows for real-time, on-site visualization; however, its application in vivo is severely limited due to insufficient depth penetration. Previous studies have highlighted the potential of using CD36-targeting antibodies as probes in photoacoustic imaging of atherosclerosis [28]. Additionally, Luo et al. demonstrated the importance of CD36-targeting peptides in fluorescence imaging for colorectal cancer, confirming the promising prospects of this method in disease monitoring [29]. Building upon these findings, our study utilizes the specific binding properties of targeting peptides, employing Indocyanine Green (ICG) as the fluorescent agent to ensure stable binding of the probe to plaque lesions. Based on this approach, we further explored its application in identifying atherosclerotic plaques and evaluating drug efficacy. Specifically, the aim of this study was to assess the feasibility of using CD36 as a target for imaging vulnerable plaque and to explore its potential for clinical diagnostic applications. Our findings suggest that the combination of the targeting peptide probe with the FLI platform offers not only high sensitivity and spatial resolution but also accurate identification of vulnerable plaque, demonstrating the significant potential for clinical application.

By leveraging the specific interaction between targeted peptides and CD36, the probe binds stably to plaque lesions, with Indocyanine Green (ICG) acting as the fluorescent agent. This approach shows significant promise for clinical application. Therefore, CD36 represents a viable target for the molecular imaging of atherosclerosis, particularly vulnerable plaque. FLI emerges as a promising technique for accurately identifying vulnerable plaque with high sensitivity and spatial resolution.

In this study, we aimed to develop a novel imaging platform for atherosclerotic plaque by constructing a molecular imaging probe that targets the lipid necrotic core using fluorescent molecules to identify high-risk plaque, based on the CD36 distribution in atherosclerosis. A new probe for active targeting was designed by conjugating a CD36 peptide with ICG (CD36pep-ICG). Following intravenous injection into atherosclerotic mice, the targeted probes specifically accumulated in plaque regions, enabling precise detection of atherosclerotic plaque. Additionally, probe-based fluorescence imaging facilitates the early diagnosis of plaque and enables monitoring of the therapeutic efficacy of anti-atherosclerotic drugs. The workflow diagram is shown in Figure 1.

## 2. Methods

### 2.1. Preparation and Characterization of CD36pep-ICG

The peptide was synthesized using the Fmoc solid-phase peptide synthesis method, with the sequence Ac-{Gly-Me}-GVIT-{Nva}-IRP (ABT-510, Qyaobio, Shanghai, China) and a purity of ≥98%. Following resin swelling, amino acid coupling, deprotection, detection, washing, and cleavage, the crude peptide was purified by high-performance liquid chromatography (HPLC). After lyophilization, the peptide was reacted with ICG-Mal via click chemistry. The resulting product was further purified by HPLC and lyophilized to yield the final product. Mass spectrometry and HPLC were employed for identification. The product was then sealed and stored at −80 °C. To investigate the stability of the fluorescence signal of the CD36pep-ICG probe in vitro, the probe was separated in a concentration gradient, and its fluorescence intensity was measured. Fluorescence images were obtained using the IVIS Lumina III imaging system (PerkinElmer, Waltham, Massachusetts, USA). To evaluate the in vitro target recognition ability of the CD36pep-ICG probe, Raw264.7 cells were seeded at 2 × 10^5^ cells per 35 mm glass-bottom dish and cultured for 24 h in normal medium or medium with 80 µg/mL ox-LDL (YB-002, yiyuan biotech, Guangzhou, China). After medium removal, cells were incubated with a 2 nM CD36pep-ICG probe for 6 h. Cells were then fixed with 4% paraformaldehyde, stained with DAPI (C1005, Beyotime, Shanghai, China), and analyzed by confocal microscopy (Thunder, Leica, Weztlar, Germany).

### 2.2. Cell Culture and Toxicity Assay

RAW 267.4 cells were cultured in DMEM supplemented with 10% FBS. Cell toxicity was evaluated using the CCK-8 assay (C0038, Beyotime, Shanghai, China). Following the protocol provided by the assay kit manufacturer and our preliminary experimental data, cells were seeded in 96-well plates at a density of 2000 cells per well and incubated at 37 °C with 5% CO₂ for 24 h. CD36pep-ICG was dissolved in DMEM containing 10% FBS at concentrations of 0, 0.5, 2.5, 5, 7.5, 10, 12.5, 15, and 20 nM. The cells were then treated with a medium containing the probes or with a control medium. After 24 h, the cells were washed three times with PBS and incubated for 2 h in 100 μL DMEM containing 10 μL of CCK-8 solution. The optical density (OD) was measured at 450 nm, and the results were expressed as a percentage of the control wells.

### 2.3. Western Blotting for CD36 Expression Detection

Mouse aortic tissue was lysed using enhanced RIPA buffer and homogenized with an ultrasonic tissue disruptor. Protein concentration was quantified using a BCA protein assay kit (P0012, Beyotime, Shanghai, China) according to the manufacturer’s protocol. The total proteins were separated by SDS-PAGE and transferred onto a 0.45 μm PVDF membrane. The membrane was blocked at room temperature with TBST buffer containing 5% non-fat milk. The proteins were then incubated with the primary antibody overnight at 4 °C. Following this, the membrane was incubated with the appropriate secondary antibody at room temperature for 120 min. Protein signals were detected using an ECL chemiluminescent solution and analyzed on the ChemiDoc system (Bio-rad, Hercules, CA, USA). Protein expression levels were quantified using Image Lab 6.1 (Bio-rad, CA, USA). GAPDH (GB15002, Servicebio, Wuhan, China) served as a loading control for normalization, including Anti-CD36 antibody (ab252922, Abcam, Cambridge, UK).

### 2.4. Animal Model

Eight-week-old male ApoE^−/−^ mice (body weight: 26 ± 2 g) and age-matched male C57BL/6 mice were obtained from Nanjing Junke Bioengineering Co., Ltd. (Jkbiot, Nanjing, China). The mice were housed in a Specific Pathogen-Free (SPF) animal facility at the Key Laboratory of Cardiovascular Disease Diagnosis and Clinical Pharmacology, Shanxi Province. Each cage housed three mice, and identification was maintained using ear tags. The animals were fed a high-fat diet with ad libitum access to food and water. The bedding was replaced every five days. All experimental procedures adhered to the guidelines for animal experimentation established by the Second Hospital of Shanxi Medical University and were approved by its Animal Ethics Committee.

### 2.5. FLI Imaging

CD36pep-ICG (20 nM, 150 µL) was administered via tail vein injection in mice. After 24 h, the mice were deeply anesthetized and euthanized. The aortas were carefully dissected and subjected to ex vivo fluorescence imaging. Imaging was conducted using the IVIS Spectrum system with an excitation wavelength of 745 nm and an emission wavelength of 840 nm. Quantitative analysis of ex vivo fluorescence intensity was performed using Living Image 3.2 (Caliper Life Sciences, Hopkinton, MA, USA).

### 2.6. Oil Red O Staining

Aortic specimens from the mice were fixed in 10% formalin for 24 h and rinsed thoroughly with normal saline to remove residual fixative. The specimens were then immersed in Oil Red O staining solution for 10–15 min to visualize lipid particles. Following staining, 75% ethanol was used for differentiation to eliminate excess dye. Finally, the specimens were mounted in glycerol gelatin and examined under a microscope to assess lipid deposition.

### 2.7. Histopathological Staining

Plaque from eight patients who underwent carotid endarterectomy was collected, and a comparative analysis was performed between the atherosclerotic core plaque and the corresponding proximal adjacent segments of the carotid artery. Human carotid endarterectomy specimens and mouse aortic valves were collected, rinsed with cold PBS, and fixed in 4% paraformaldehyde for over 12 h. The specimens were embedded in paraffin, sectioned into 4–5 slices at a thickness of 4 μm, and stained using fluorescence techniques. Histological changes were examined under a fluorescence microscope.

## 3. Result

Aberrant CD36 expression in the atherosclerotic plaque of patients

We analyzed single-cell RNA sequencing data (GSE159677) from post-operative carotid endarterectomy (CEA) samples, consisting of three atherosclerotic core (AC) plaque and three patient-matched proximal adjacent (PA) portions of carotid artery tissue. The cells were categorized into 15 main types (Appendix A). Cell proportion analysis revealed a significant difference in macrophage-derived foam (Foamy Mac) cells between the two groups, with the proportion of foamy Mac significantly increased in the AC group (Figure 2a). KEGG pathway analysis indicated that lipid metabolism plays a key role in the development and progression of atherosclerotic plaque (Appendix A). We further focused on differentially expressed genes associated with lipid metabolism and plaque core regions. Among these, CD36, an ox-LDL uptake receptor, was identified as a key gene differentiating the control and plaque groups. CD36 expression was significantly elevated in foam cells located within the plaque core (Figure 2b). Additionally, CD36 expression was markedly higher in CEA postoperative samples, whereas its expression intensity significantly decreased in remote plaque controls (Figure 2c,d). These findings confirm that CD36 serves as a potential marker and therapeutic target for lipid metabolism regulation in atherosclerotic plaque.

2.Lipid Metabolism Is Dysregulated in the Early Stage of Plaque

To investigate the critical role of lipid metabolism in atherosclerosis progression and the potential alteration of CD36 in atherosclerosis lesions, we employed a high-fat diet (HFD)-induced ApoE^−/−^ mouse model of atherosclerosis. Blood lipid analysis revealed significant differences in the levels of total cholesterol (TC), triglycerides (TG), and low-density lipoprotein (LDL-c), indicating a lipid disorder in the mice that mirrored the pathological characteristics of atherosclerosis (Figure 3a). The level of CD36 protein was measured in mouse aorta samples, and compared with the control group, the expression of CD36 in aortic atherosclerotic plaque tissues was significantly upregulated (Figure 3b,c), which was consistent with its role in ox-LDL uptake and foam cell formation as a scavenger receptor in atherosclerosis. The elevated expression of CD36 is associated with lipid accumulation and increased inflammation within the plaque. These findings confirm that CD36 plays a pivotal role in lipid uptake and metabolism in plaque. Furthermore, lipid metabolism disorders appear early in atherosclerosis, with CD36 expression positively correlated with lipid dysregulation, increasing as the disease progresses. These results provide a solid biological foundation for the future development of diagnostic and therapeutic strategies targeting CD36, as well as further research into the CD36pep-ICG probe, which specifically binds to diseased cells.

3.Synthesis and characterization of fluorescent imaging probes

CD36pep-ICG was synthesized using a self-synthesized CD36 polypeptide and modified with ICG. Fluorescence spectroscopy was performed (Figure 4c), with an excitation wavelength (Ex) set to 740 nm and the emission spectrum (Em) measured between 750 nm and 850 nm. The sample exhibited the highest fluorescence intensity at 820 nm, consistent with the known fluorescence properties of ICG. This result confirms the presence of ICG in the sample and indicates that its fluorescence properties remain stable during coupling, providing a basis for further quantitative analysis and potential applications in disease imaging studies. Fluorescence imaging signals were detected at CD36pep-ICG probe concentrations of 0, 0.625, 1.25, 2.5, 5, 10, and 20 nM. The fluorescence intensity of CD36pep-ICG demonstrated a linear positive correlation with probe concentration in the range of 0–20 nM (Figure 4a), with an R^2^ value of 0.9956 (Figure 4b). Different concentrations of CD36pep-ICG were incubated with mouse macrophages (Raw264.7), and cell viability was assessed after 6 h. CCK-8 assays revealed no significant cytotoxic effects (Appendix A). To accurately evaluate the targeted recognition ability of the CD36pep-ICG on macrophage-derived foam cells, Raw264.7 cells treated with ox-LDL were co-incubated with the probe, and imaging was conducted using confocal laser microscopy. The results indicated that, compared to the control group, the CD36pep-ICG probe was specifically bound to the diseased cells through the high expression of CD36, generating a prominent green fluorescence signal (Figure 4d). These results collectively support the potential application of CD36pep-ICG as a valuable tool for molecular imaging and further investigation of lipid metabolism in atherosclerotic lesions.

4.FLI imaging based on CD36pep-ICG probe predicts the occurrence of atherosclerosis

As shown in Figure 5a, fluorescence imaging was conducted on mice from the normal diet (ND), 2-month high-fat diet (HFD 2 m), and 5-month high-fat diet (HFD 5 m) groups after probe injection to assess the correlation between signal intensity and lesion severity. The mice were deeply anesthetized and euthanized, and their aortas were carefully dissected for ex vivo fluorescence imaging and Oil Red O staining. The results revealed that compared to the ND group, probe accumulation, and FLI signals were more pronounced in the aortic lesions of early-stage HFD mice, with an even stronger FLI signal observed in ApoE^−/−^ mice after 5 months of high-fat feeding (Figure 5b). The results of ex vivo Oil Red O staining further confirmed that the FLI fluorescence signals corresponded to the lesion areas. Classic Oil Red O staining of the mouse aortas showed significant changes in lipid droplet area in ApoE^−/−^ mice after 2 months of high-fat feeding compared to controls. After 5 months of high-fat feeding, lipid droplet accumulation was more prominent. Statistical analysis of the ex vivo aortic Oil Red O area and FLI signal values, shown in Figure 5d, indicated that high-intensity fluorescence signals were obtained from the aortic plaque regions of HFD ApoE^−/−^ mice. Ex vivo FLI imaging demonstrated that FLI signal intensity increased with the severity of aortic lesions. Immunofluorescence results showed that CD36 expression was significantly correlated with plaque lesions in both early and late-stage atherosclerotic models (Figure 5c). As a scavenger receptor, CD36 is highly expressed in atherosclerotic plaque, with its fluorescence signals predominantly concentrated in lipid-rich regions. Figure 5e presents the semi-quantitative fluorescence results. These findings suggest that the CD36pep-ICG exhibits strong plaque-targeting properties, can detect early aortic lesions, predict the onset of atherosclerosis, and facilitate early intervention.

5.The CD36pep-ICG probe is used for drug efficacy monitoring

Early detection of asymptomatic atherosclerotic plaque, combined with appropriate drug therapy, can significantly reduce the incidence of cardio-cerebral vascular events. Due to its strong imaging capabilities and biocompatibility, we explored the application of CD36pep-ICG for the early diagnosis of atherosclerosis and the evaluation of anti-atherosclerotic drug efficacy (Figure 6a). Atorvastatin (AT), a well-established anti-atherosclerotic drug, treats atherosclerosis by inhibiting HMG-CoA reductase in the liver, thereby reducing cholesterol synthesis and lowering LDL-C, TC, and TG levels. To develop an early-stage atherosclerotic mouse model, ApoE^−/−^ mice were fed a high-fat diet for 2 months. During the feeding period, the mice were divided into two groups: one received no drug treatment (HFD 2m), and the other was treated with AT via tail vein injection at a dose of 10 mg/kg/day (HFD 2m + AT). Mice fed a normal diet (ND group) served as controls. After probe injection, aortas were collected for analysis 24 h later. As shown in Figure 6b, the HFD 2m group without AT treatment exhibited significantly enhanced FLI signals compared to the control group. This result demonstrates that the CD36pep-ICG probe can detect early-stage plaque in ApoE^−/−^ mice fed an HFD for 2 months with high sensitivity and contrast. Additionally, the FLI signals in the aorta closely corresponded to Oil Red O-positive staining regions, confirming that the CD36pep-ICG probe specifically targeted atherosclerotic areas. AT treatment significantly reduced both the fluorescence imaging signals and Oil Red O-positive staining areas in the aorta compared to the untreated HFD 2m group (Figure 6d). Pathological analysis of the aortic valve further revealed early lipid accumulation in ApoE^−/−^ mice fed an HFD for 2 months, while AT treatment significantly reduced the CD36 fluorescence signals (Figure 6c). In conclusion, CD36pep-ICG-based FLI imaging enables the early detection of aortic lesions and provides a reliable method for monitoring and evaluating the therapeutic efficacy of anti-atherosclerotic drugs.

## 4. Discussion

Lipid-driven rupture of atherosclerotic plaque is a leading cause of myocardial infarction and ischemic stroke in developed countries [30,31]. Effectively assessing and predicting cardiovascular risk remains a critical challenge in modern cardiovascular medicine. Traditional assessment methods, such as coronary artery calcification scores, coronary computed tomography angiography, carotid intima-media thickness, and the ankle-brachial index, primarily rely on anatomical imaging [32,33]. However, these methods focus on quantifying plaque burden and have limited capacity to evaluate plaque biology, particularly stability and vulnerability [34,35]. With advancements in personalized medicine, non-invasive molecular imaging has garnered increasing attention for its ability to identify atherosclerotic plaque and assess its biological characteristics [26,36,37]. In this study, we developed a novel molecular probe, CD36pep-ICG, designed to identify vulnerable atherosclerotic plaque by targeting the CD36 protein, offering new potential for early diagnosis and treatment.

The effective application of molecular imaging technology relies on the precise selection of appropriate molecular targets and imaging platforms. In atherosclerosis research, CD36 signaling has been increasingly recognized for its key role in promoting plaque inflammation. Using an ApoE⁻/⁻ mouse model, previous studies demonstrated that atherosclerotic lesion size was reduced while a stable plaque phenotype was induced, characterized by a limited abundance of inflammatory cells and thickening of the fibrous cap. Furthermore, inhibiting the interaction between CD36 and its downstream molecule, TNF receptor-associated factor 6 (TRAF6) almost completely halted the progression of atherosclerosis [25]. The absence of CD36 in mouse endothelium further confirmed that inflammatory cell recruitment to the plaque was reduced, resulting in the formation of more stable plaque [26]. In our study, we observed that CD36 expression was significantly upregulated in aortic tissue in the high-fat diet group. Therefore, the elevated expression of CD36 holds promise as an imaging target for detecting vulnerable plaque.

Several studies have investigated fluorescent probes targeting various biomarkers for plaque detection, including those aimed at matrix metalloproteinases (MMPs) and integrins. MMP-targeting probes, such as those targeting MMP-2/9, have shown promise in identifying vulnerable plaque, as MMP activity is elevated in areas of plaque rupture [38,39]. Similarly, integrin-targeted probes have been developed for non-invasive imaging of endothelia l activation, a hallmark of plaque instability [40,41]. However, these probes often suffer from limitations in both specificity and sensitivity, particularly in early plaque detection. In contrast, our CD36pep-ICG probe specifically targets CD36, a protein highly expressed in inflammatory cells within atherosclerotic plaque, suggesting its potential for greater specificity in identifying vulnerable plaque regions.

Optical imaging offers the advantages of high sensitivity and simple operation. Combining optical imaging with precision molecular targets holds significant potential for achieving high-precision, high-sensitivity imaging of atherosclerotic plaque. Molecular imaging employs two main strategies: passive targeting and active targeting. Passive targeting relies on the enhanced permeability and retention effect of blood vessels, whereas active targeting involves binding contrast agents to molecular targets on the surface of specific cells or tissues. This approach facilitates the accumulation of contrast agents at targeted sites, thereby enhancing imaging contrast. In this study, the combination of ICG and CD36 peptides enables the identification of molecular characteristics of plaque, improving early detection sensitivity and diagnostic accuracy. Recent studies suggest that combining fluorescent probes with multimodal imaging platforms may improve diagnostic accuracy. For example, the combination of optical imaging with positron emission tomography (PET) or MRI has been explored in atherosclerosis imaging to overcome the depth penetration limitations of fluorescence-based techniques [42,43,44]. In our study, the use of optical imaging with ICG, in conjunction with the targeting ability of CD36 peptides, can be adapted for multimodal imaging, further enhancing its clinical application. While other fluorescent probes have shown promise in atherosclerotic plaque imaging, the specificity of CD36-targeting peptides combined with the high sensitivity of ICG makes our probe a promising candidate for the early detection of vulnerable plaques. Future studies comparing the CD36pep-ICG probe with other fluorescent probes in vivo will be essential to fully assess its diagnostic potential.

The current study has several limitations, including the absence of in vitro binding assays, which are essential for directly confirming the specificity of the probe and evaluating its binding affinity. Although the primary focus of our research is on imaging in a mouse model and ex vivo validation, we acknowledge the importance of incorporating additional cell-based experiments to achieve a more comprehensive understanding of the probe’s performance. Consequently, future studies will integrate in vitro assays to further assess the probe’s specificity and binding affinity. Although this study shows the specific accumulation of the CD36pep-ICG probe in aortic atherosclerotic plaque, it does not provide direct evidence of CD36pep binding to specific sites within the plaque. Therefore, further experiments are needed to confirm whether CD36pep specifically targets CD36 expression sites. Additionally, due to the limited depth penetration of fluorescence detection, this study primarily focuses on ex vivo validation. Future research aims to enhance imaging strategies by incorporating novel techniques and a probe, such as magnetic particle imaging (MPI) and photoacoustic imaging (PAI). These advancements could facilitate non-invasive in vivo imaging in mice and, ultimately, enable clinical applications. Furthermore, we plan to incorporate control probes with scrambled peptides in future studies to conduct additional experiments, helping to exclude interference from non-specific phagocytosis and better assess the signal’s specificity.

## Figures and Tables

**Figure 1 pharmaceutics-17-00444-f001:**
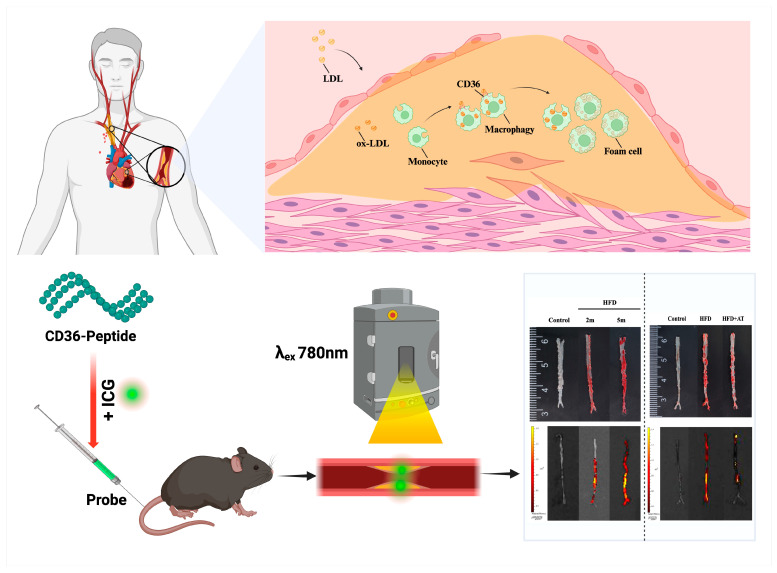
Schematic diagram of CD36pep-ICG with CD36 targeting function for vulnerable plaque theranostic. Macrophages phagocytose ox-LDL via the CD36 receptor, leading to foam cell formation and the development of vulnerable plaque, thereby confirming the critical role of CD36 in plaque formation. Based on this mechanism, CD36pep-ICG probes were synthesized to detect vulnerable plaque and assess drug efficacy using ex vivo fluorescence imaging and pathological detection in mice.

**Figure 2 pharmaceutics-17-00444-f002:**
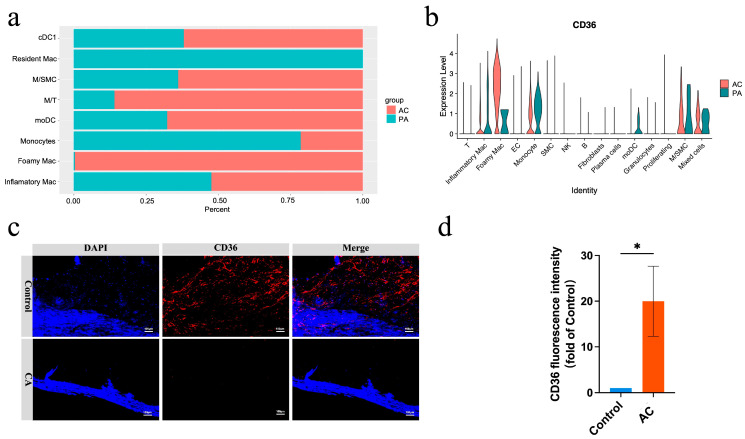
(**a**) Cell number and proportion of macrophage subsets in AC and PA. (**b**) Expression of CD36 in cell distribution of AC and PA. (**c**) Immunofluorescence images of CD36 in AC and PA. (**d**) Quantification analysis of CD36 immunofluorescence in AC and PA. (*n* = 3, * *p* < 0.05).

**Figure 3 pharmaceutics-17-00444-f003:**
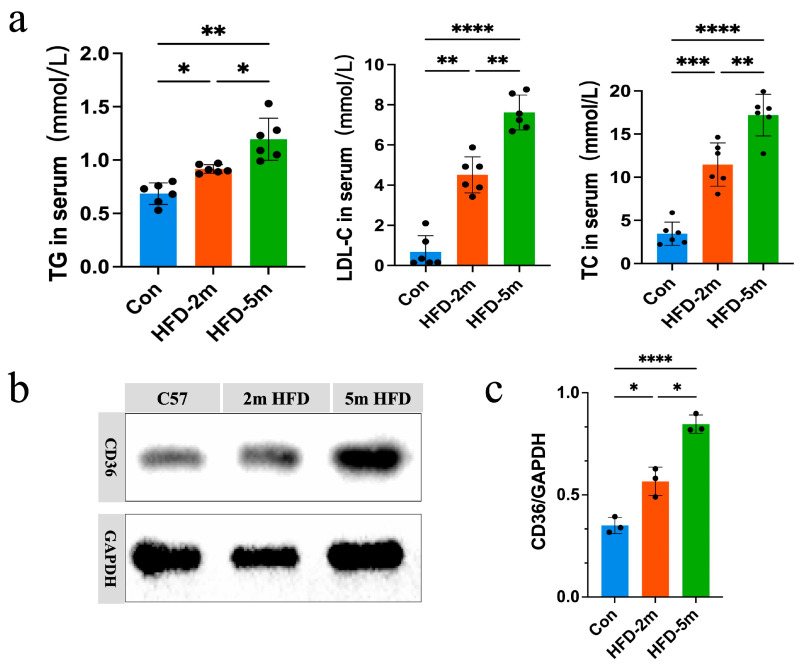
(**a**) TG, LDL-C, and TC levels in the mouse serum of different experimental groups (*n* = 6). (**b**,**c**) Expression levels of CD36 in the aorta of different experimental groups. The relative expression level was statistically different (*n* = 3). All data are shown as mean ± SD. (* *p* < 0.05, ** *p* < 0.01, *** *p* < 0.001, **** *p* < 0.0001).

**Figure 4 pharmaceutics-17-00444-f004:**
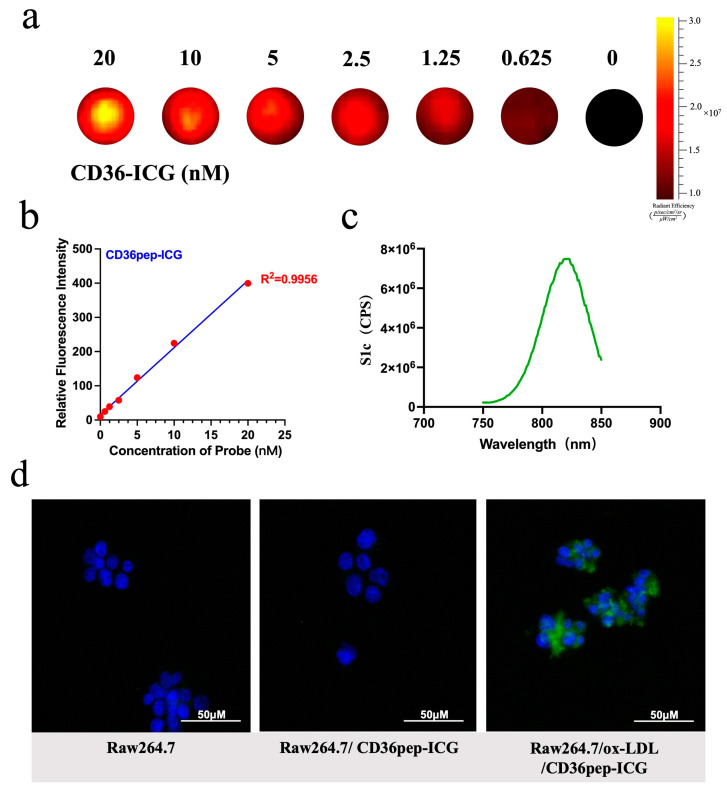
(**a**,**b**) CD36pep-ICG characterization tests. Standard curve generated by quantifying fluorescence intensity of various concentrations of CD36pep-ICG (concentrations in nM). R^2^ = 0.9956. (**c**) Emission spectra of CD36pep-ICG. (**d**) Confocal fluorescence images of ox-LDL-induced and non-induced Raw264.7 cells treated with CD36pep-ICG.

**Figure 5 pharmaceutics-17-00444-f005:**
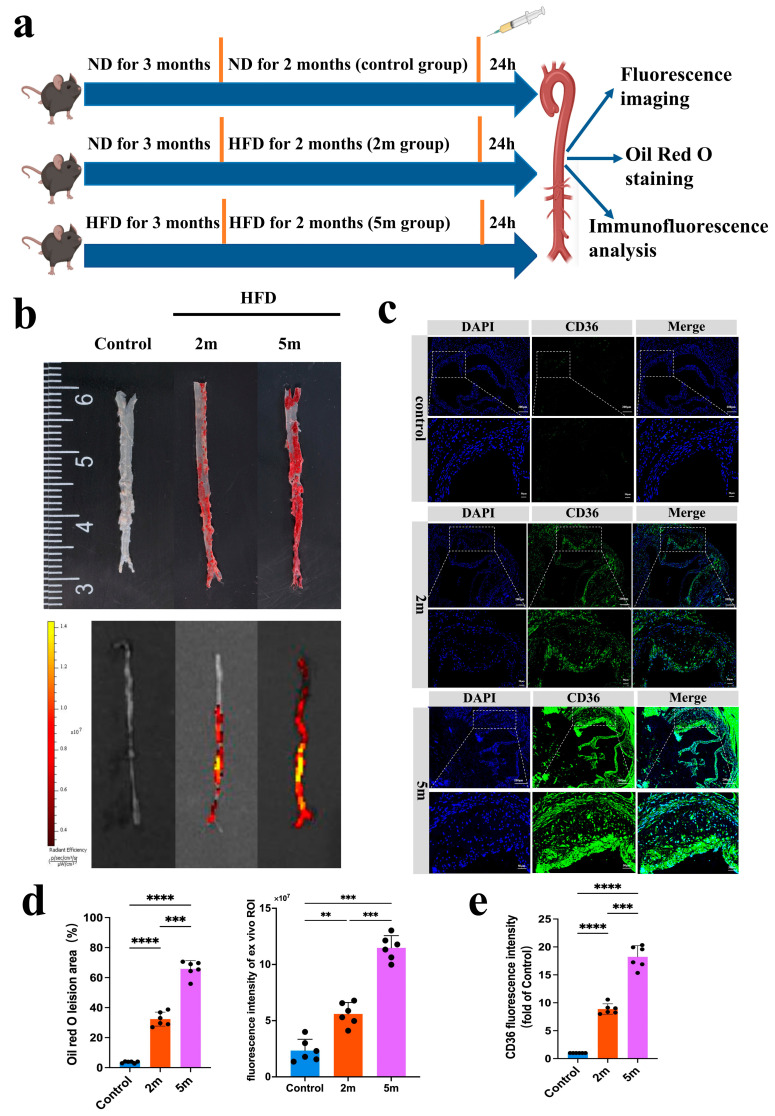
(**a**) The diagram illustrates the design of a monitoring experiment to evaluate the effects of CD36pep-ICG on different degrees of atherosclerosis. The C57BL/6 mice were fed with a normal diet for 5 months and served as the control group (control group). The ApoE^−/−^ mice were fed with an HFD for just 2 months to develop early-stage plaque (2m group). The ApoE^−/−^ mice were fed with an HFD for 5 months to develop advanced-stage plaque (5m group). At the end of the feeding period, mice in each group were administered CD36pep-ICG, and their whole aortas were harvested for analysis 24 h after the probe injection. (**b**) Representative FL images and Oil Red O staining images of the same whole aorta isolated from the Control, 2m group, and 5m group. The three groups of mice were intravenously injected with a CD36pep-ICG probe. The mice were sacrificed at 24 h postinjection and the whole aortas were imaged. (**c**) Immunofluorescence images of CD36 in aortic valve from Control, 2m group, and 5m group. The blue color represents DAPI staining and the green color indicates CD36. (**d**) Quantification analysis of Oil Red O staining and FL intensity in (**b**). (**e**) Quantification analysis of CD36 immunofluorescence in (**c**). (** *p* < 0.005, *** *p* < 0.001, **** *p* < 0.0001).

**Figure 6 pharmaceutics-17-00444-f006:**
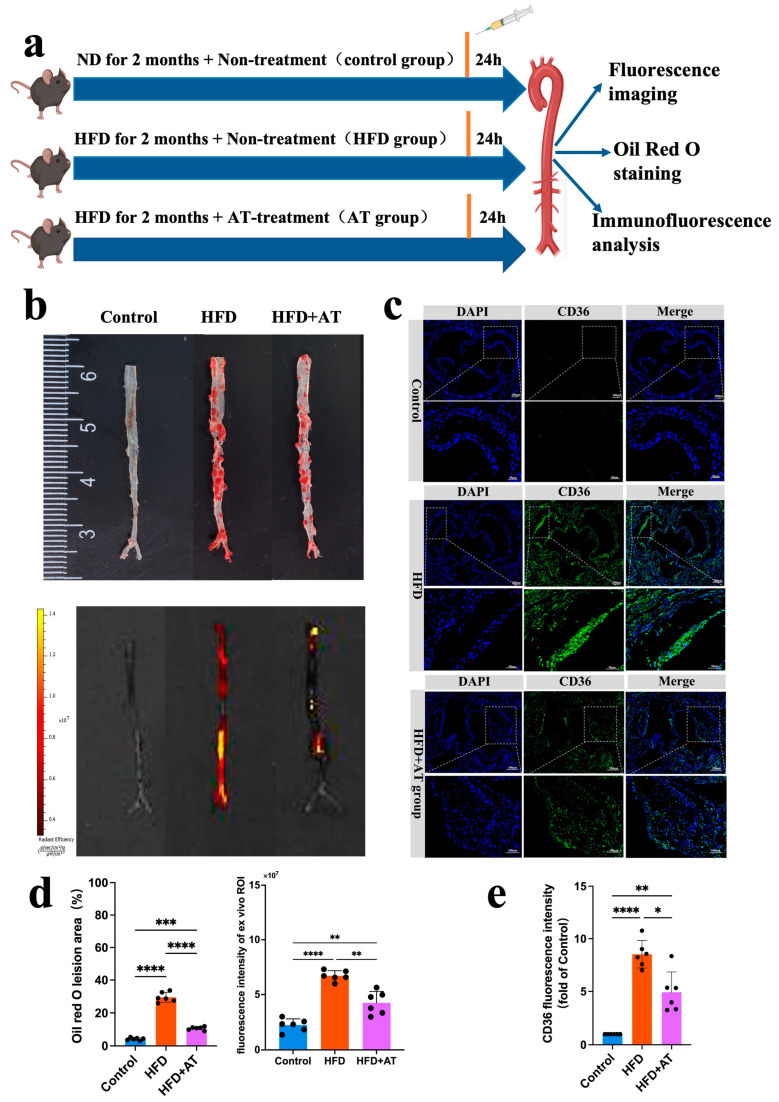
(**a**) A schematic showing the experimental design to assess the effects of CD36pep-ICG in the early detection of atherosclerosis and monitoring of anti-atherosclerotic drug efficacy. The diagram illustrates the design of a monitoring experiment to evaluate the effects of CD36pep-ICG on different degrees of atherosclerosis. The ApoE^−/−^ mice were fed with a normal diet for 2 months and served as the control group (control group). The ApoE^−/−^ mice were fed with an HFD for just 2 months to develop early-stage plaque. During HFD feeding, one group of mice was treated with AT drug (10 mg kg−1 per day for two months) (HFD + AT group). The other group of mice received no treatment (HFD group). (**b**) Representative FL images and Oil Red O staining images of the same whole aorta isolated from Control, HFD group, and HFD + AT group. The three groups of mice were intravenously injected with the CD36pep-ICG probe. The mice were sacrificed at 24 h postinjection and the whole aortas were imaged. (**c**) Immunofluorescence images of CD36 in aortic valve from the Control, HFD group, and HFD + AT group. (**d**) Quantification analysis of Oil Red O staining and FL intensity in (**b**). (**e**) Quantification analysis of CD36 immunofluorescence in (**c**). (*: *p* < 0.05, ** *p* < 0.005, *** *p* < 0.001, **** *p* < 0.0001).

## Data Availability

The raw data supporting the conclusions of this article will be made available by the authors, without undue reservation, to any qualified researcher.

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
