# Peer review of "Advanced Detection and Therapeutic Monitoring of Atherosclerotic Plaque Using CD36-Targeted Lipid Core Probe"

_pharmaceutics, 2025, doi:10.3390/pharmaceutics17040444_

Round 1
Reviewer 1 Report
Comments and Suggestions for Authors
In the article entitled “Advanced Detection and Therapeutic Monitoring of 2 Atherosclerotic Plaque Using CD36-Targeted Lipid Necrosis Probe” the main objective of the authors is to produce and evaluated an imaging probe targeting CD36 dedicated to the fluorescent imaging od vulnerable atherosclerotic lesions. The rationale being that CD36, a scavenger receptor, plays a key role in atherosclerotic lesions. Its expression by macrophages support the hypothesis that it could be use as a target for molecular imaging of inflamed lesions that are responsible for acute events. The rational and target selection are therefore pertinent. The role of CD36 in human and mice atherosclerotic lesion being well known, the novelty of his study therefore mainly relies on the development and evaluation of the imaging probe.
While the study design allows confirming the expression of CD36 in both human and mice lesions, it lacks some important elements regarding the evaluation of the imaging agent. I would therefore recommend to the authors to strengthen this part of the study.
· Line 65-78: I kindly disagree with the authors concerning the fact that various imaging modalities are “widely used in clinical practice to detect atherosclerotic plaques”. Different modalities are being investigated but none is use in standard care. I am also not certain that patient cooperation to avoid image blurring constitute a strong limitation to MRI, and I would suggest to mention that CCTA that is (to my knowledge) the most accurate modality for the detection of hypodense necrotic and lipidic core (that is also targeted in the present study) (see for example the scot-heart study).
· In the introduction section, there is a bit of redundancy in between lines 87-92 and the following paragraph. I was not clear to me which sentence referred to previous work and wich one concerned the present work. I would therefore suggest to more clearly state what are the preliminary results and then the objective and main finding of the present study.
· While the authors mention limitation of other imaging modalities in the introduction and discussion sections, they do not discuss the advantages and limitations of their approach. For instance, the present study focus on ex vivo measurements, but I assume that the final objective is to perform in vivo imaging in mice and then in the clinic. The choice of an invasive technique, that is not necessarily convenient to screen a large patient population, should be discussed.
· The abstract is somehow misleading : It reads line 22-23 that “ In vivo imaging in atherosclerotic mouse models confirmed selective probe accumulation in plaque and its potential to monitor disease progression.” While there is no in vivo imaging in this study (and no demonstration of he selectiveness of the probe). Similarly, the term “drugs” l25 should be singular.
· Please state in the method section (l165) how many human carotid endarterectomy samples were collected, and what was the criteria that you used to allocate them into the 2 groups.
· Some methods and results are meant to confirm previous knowledge (Expression of CD36 in human and in mice atherosclerosis for example) while other are dedicated on investigating new hypothesis (the evaluation of the imaging probe). While it is a very good point to confirm previously knowledge and validate the animal model, I believe that the authors should discriminate more between the results that confirm a previous knowledge and the one that present new information. Indeed, it sometimes reads a bit as if the expression of CD36 in human/mice atheroma has not been reported previously (as it is acknowledged by the authors in the discussion l326-337). For example line 184 it reads “These findings suggest that CD36 serves as a potential marker and therapeutic target for lipid metabolism regulation in atherosclerotic plaque.” And line 202 that “These findings suggest that CD36 plays a pivotal role in lipid uptake and metabolism in plaque.” The term “confirm” for example could be prefer to “suggest” in the later.
· I am confused by the use of the term destroyed l191. What exactly do the authors means by that? Dysregulation?
· Figure 3b: The image of CD36 is similar to that provided in raw supplemental data while that of GAPDH is reversed (left-right) suggesting either that the figure is wrong or that the CD36/GAPDH ratios were calculate using the incorrect lanes. Please check.
· While the production and evaluation of the imaging agent constitute the main novelty in this study, it lacks important data:
o No in vitro binding assay is presented to confirm the specificity of the probe or determine the KD.
o No measurement of the autofluorescence of the artery is mentioned. Plaques contain material such as lipids, cholesterol crystal and fibrosis that can be responsible for elevated autofluorescence. It is very important to measure it.
o No study was dedicated to the evaluation of the specificity of the signal while it is stated that the signal is specific at multiple occasions throughout the manuscript. A nonspecific uptake (not related to CD36 expression) could occur, for example a binding to the lipid core. An in vivo competition study could have been set up for example, or a control probe with scramble peptides employed a negative control.
· Symbols showing the significance level (*, **) are missing from graphics of figure 5 and 6.
· The discussion is short and mainly summarize the rational and results, with only limited comparisons of the obtained results with that obtained by others. The addition of a paragraph in the discussion section comparing the results to those obtained with other fluorescent probes for instance would I believe greatly enhance the manuscript.
· I am not convinced that the term “lipid necrosis probe” is well suited in the title. While there is some CD36 within the cellular debris of the necrotic core, it is not a target per se of necrosis.
Comments on the Quality of English LanguageMy own english is far from perfect. I had no difficulties to read and understand the manuscript. However some terms could be improved (such as destroyed lipid metabolism)
Author Response
Many thanks to you for your careful review and a series of sincere and useful suggestions for our manuscript. Here is a point-to-point response for all your concerns and we hope that we have addressed all of your concerns.
Changes within the manuscript we will identify with a yellow background color.

Reviewer 2 Report
Comments and Suggestions for Authors
This study aims to develop a novel molecular imaging platform for detecting atherosclerotic plaque using a fluorescent probe targeting lipid necrotic cores.
I have several comments regarding this manuscript:
1. I recommend to use the full term atherosclerosis instead of AS abbreviation
2. Please provie the Name of the CCK-8 assay vendor in 2.2 Section. Why did you use 2000 cells for CCK-8 assay?
3. The quality and resolution of the Fig. 2a must be improved.
4. The quality of the Fig. 4a must be improved (indicate units and Control in the Fig captions etc).
5. There is no data, illustrating CD36pep specific accumulation within plaques.
6. Caption of the figure 1 should be improved with more detail explanation of the technology (vital imaging of plaque or not, etc.)
Author Response

(The authors gave the same response as above.)

Round 2
Reviewer 1 Report
Comments and Suggestions for Authors
I would like to thank the authors for their answers. All the questions that did not required additional experiments have been well addressed. I remain however convinced that the article lacks crucial experiments regarding the in vitro and in vivo characterization of the imaging probe.
From my point of view, it is not possible to conclude from the results that the probe is validated while not providing any evidence that it does actually bind to its target, at least in vitro, and that the signal is specific. The authors have acknowledged this point, but while it is now stated in a limitation section, no additional experiment has been conducted to address it.
Author Response
Many thanks to you for your careful review and a series of sincere and useful suggestions for our manuscript. Here is a point-to-point response for all your concerns and we hope that we have addressed all of your concerns.

Round 3
Reviewer 1 Report
Comments and Suggestions for Authors
The authors have provided an additionnal an result regarding the in vitro binding of their probe to macrophages stimulated with oxLDL. There is however no source for these results and they have not been added to the manuscript. Can the author either clearly refer to this previously published data in their article, or include them in their manuscript?
Author Response

(The authors gave the same response as above.)
